# Searching for Low-Bit Weights in Quantized Neural Networks

**Zhaohui Yang**[1,2], **Yunhe Wang**[2], **Kai Han**[2], **Chunjing Xu**[2],
**Chao Xu**[1], **Dacheng Tao**[3],[*] **Chang Xu**[3]
[1] Key Lab of Machine Perception (MOE), Dept. of Machine Intelligence, Peking University.
[2] Noah's Ark Lab, Huawei Technologies.
[3] School of Computer Science, Faculty of Engineering, University of Sydney.
`zhaohuiyang@pku.edu.cn;` `{yunhe.wang,kai.han,xuchunjing}@huawei.com`
`xuchao@cis.pku.edu.cn;` `{dacheng.tao, c.xu}@sydney.edu.au`

## Abstract

Quantized neural networks with low-bit weights and activations are attractive for developing AI accelerators. However, the quantization functions used in most conventional quantization methods are non-differentiable, which increases the optimization difficulty of quantized networks. Compared with full-precision parameters (*i.e.*, 32-bit floating numbers), low-bit values are selected from a much smaller set. For example, there are only 16 possibilities in 4-bit space. Thus, we present to regard the discrete weights in an arbitrary quantized neural network as searchable variables, and utilize a differential method to search them accurately. In particular, each weight is represented as a probability distribution over the discrete value set. The probabilities are optimized during training and the values with the highest probability are selected to establish the desired quantized network. Experimental results on benchmarks demonstrate that the proposed method is able to produce quantized neural networks with higher performance over the state-of-the-art methods on both image classification and super-resolution tasks. The PyTorch code will be made available at `https://github.com/huawei-noah/Binary-Neural-Networks/tree/main/SLB` and the MindSpore code will be made available at `https://www.mindspore.cn/resources/hub`.

## 1 Introduction

The huge success of deep learning is well demonstrated in considerable computer vision tasks, including image recognition [27, 61, 62], object detection [15, 53], visual segmentation [25], and image processing [35, 72]. On the other side, these deep neural architectures are often oversized for accuracy reason. A great number of network compression and acceleration methods have been proposed to eliminate the redundancy and explore the efficiency in neural networks , including pruning [23, 67], distillation [69, 6, 70, 44, 14], low-bit quantization [8, 52, 78], weight decomposition [75, 42, 71], neural architecture search [46, 68] and efficient block design [30, 54, 50, 21, 64].

Among these algorithms, quantization is very particular which represents parameters in deep neural networks as low-bit values. Since the low costs of quantized networks on both memory usage and computation, they can be easily deployed on mobile devices with specific hardware design. For example, compared with conventional 32-bit networks, binary neural networks (BNNs) can directly obtain a $32\times$ compression ratio, and an extreme computational complexity reduction by executing

---

[*]Corresponding Author.

bit-wise operations (*e.g.*, 57× speed-up ratio in XNORNet [52]). However, the performance of the low-bit neural networks is usually worse than that of full-precision baselines, due to the optimization difficulty raised by the low-bit quantization functions.

To reduce the accuracy drop of quantized neural networks, some methods have been proposed in recent years. BiRealNet [48] inserts more shortcuts to help optimization. Structured Binary [81] and ABCNet [43] explore some sophisticated binary blocks and achieve comparable performance to that of full-precision networks, *etc*.

Admittedly, the aforementioned methods have made great efforts to improve the performance of the quantized neural networks. However, the accuracy gap between the full-precision network and its quantized version is still very huge, especially the binarized model. For example, the state-of-the-art accuracy of binarized ResNet-18 is about 10% lower than that of the full-precision baseline. A common problem in existing quantization methods is the estimated gradients for quantization functions, using either STE [8, 52, 29, 47, 59, 4] or self-designed gradient computation manner [46]. The estimated gradients may provide inaccurate optimization direction and consequently lead to worse performance. Therefore, an effective approach for learning quantized neural networks without estimated gradients is urgently required.

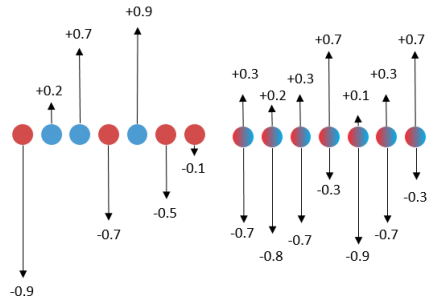

In this paper, we present a novel weight searching method for training the quantized deep neural network without gradient estimation. Since there are only a few values of low-bit weights (*e.g.*, +1 and -1 in binary networks), we develop a weight searching algorithm to avoid the non-differentiable problem of quantization functions. All low-bit values of an arbitrary weight in the given network are preserved with different probabilities. These probabilities will be optimized during the training phase. To further eliminate the performance gap after searching, we explore the temperature factor and a state batch normalization for seeking consistency in both training and testing. The effectiveness of the proposed differentiable optimization method is then verified on image classification and super-resolution tasks, in terms of accuracy and PSNR values.

Figure 1: The binary case of the proposed method SLB. Conventional BNN (left) binarize full-precision proxy weights during training and update proxy weights by inaccurate gradients. For SLB (right), the training period is to optimize the distribution over discrete weights and the specific binary values will be determined after training.

## 2    Related Works

To reduce memory usage and computational complexity of deep neural networks, a series of methods are explored, including efficient block design, network pruning, weight decomposition, and network quantization. MobileNet [30, 54] and ShuffleNet [50] design novel blocks and construct the networks with high efficiency. Weight pruning methods such as DeepCompression [23] propose to remove redundant neurons in pre-trained models, while requires specific hardware for acceleration. Structured pruning methods [28, 67, 73] discard a group of weights (*e.g.*, an entire filter) in pre-trained networks so that the compressed model can be directly accelerated in any off-the-shelf platforms. Lxow-rank decomposition methods [10, 34, 75, 71] explore the relationship between weights and filters and explore more compact representations. Gradient-based neural architecture search methods [46, 66, 20, 60, 38, 39] use the continuous relaxation strategy to search discrete operations for each layer. All the candidate operations are mixed with a learnable distribution during searching and the operation with the highest probability is selected as the final operation. Quantization methods [40, 1, 51, 16, 11, 65, 77, 22] represent weights or activations in a given neural architecture as low-bit values. The model size and computational complexity of the quantized network are much smaller than those of its original version. Compared with previous model compression methods, network quantization does not change the architecture and the calculation process of original ones, which is supported by some mainstream platforms.

To maintain the high performance of quantized neural networks, considerable approaches have been explored for training low-bit weights and activations. For instance, Fixed point quantization [41]

reduces about 20% complexity without loss of accuracy on CIFAR-10 dataset. BinaryConnect [8] represents weights by binary values $\{-1, +1\}$, TTQ [79] uses the ternary weights $\{-1, 0, +1\}$. Activations in these two methods are still full-precision values. DoReFa [78] quantizes both weights and activations to low-bit values. Among all the quantization methods, in extreme cases, if activations and weights are both 1 bit, the binary neural network not only occupies less storage space but also utilizes low-level bit operations xnor and bitcount to accelerate inference. ABCNet [43], CBCN [45], BENN [80] and SBNN [81] explored how to modify the BNN to match the performance with full-precision networks. In [82], a clearly different viewpoint is investigated into gradient descent, which provides the first attempt for the field and is worth further exploration. The hidden relationship between different variables is carefully studied, which will benefit many applications.

Nevertheless, most of the quantization functions in existing methods are non-differentiable, which increases the optimization difficulty. Thus, this paper aims to explore a more effective algorithm for quantizing deep neural networks.

# 3 Methods

In this section, we first briefly introduce the optimization approach in the conventional weight quantization methods and the existing problems. Then, a low-bit weight search approach is developed for accurately learning quantized deep neural networks. The strategies, including gradually decreasing temperature and state batch normalization, are proposed to eliminate the quantization gap.

## 3.1 Optimization Difficulty in Network Quantization

Representing massive weights and activations in deep neural networks using low-bit values is effective for reducing memory usage and computational complexity. However, it is very hard to learn these low-bit weights in practice.

For the case of $q$-bit quantization, the coordinate axis is divided into $m = 2^q$ intervals, and each interval corresponds to one discrete value in set $\mathbb{V} = \{v_1, v_2, \cdots, v_m\}, v_1 < \cdots < v_m$. Denoting the thresholdsas $T = \{t_1, \ldots, t_{m-1}\}, v_1 < t_1 \leq v_2 \ldots v_{m-1} < t_{m-1} \leq v_m$, the quantization function $\mathcal{Q}(x)$ can be defined as,

$$\mathcal{Q}(x) = \begin{cases} v_1 & \text{if } x < t_1, \\ v_2 & \text{if } t_1 \leq x < t_2, \\ \cdots \\ v_m & \text{if } t_{m-1} \leq x, \end{cases} \tag{1}$$

where $x \in \mathbb{R}$ is the original full-precision values. Given full-precision latent convolutional filters $W_q^{lat}$, the weights are quantized as $W_q = \mathcal{Q}(W_q^{lat})$, which are used to calculate the output features (*i.e.*, activations). In the training phase, the gradients $\nabla_{W_q}$ w.r.t the quantized filters $W_q$ can be calculated with standard back-propagation algorithm. However, the gradients $\nabla_{W_q^{lat}}$ is hard to obtain, because the derivative of $\mathcal{Q}$ is zero almost everywhere and infinite at zero point. The Straight Through Estimator (STE) [8] is a widely used strategy to estimate the gradient of $W_q^{lat}$, *i.e.*,

$$\nabla_{W_q^{lat}} = \nabla_{W_q} \tag{2}$$

where the approximate gradients $\nabla_{W_q^{lat}}$ are use to update weights $W_q^{lat}$,

$$\hat{W_q^{lat}} = W_q^{lat} - \eta \cdot \nabla_{W_q^{lat}} \cdot \sigma(\nabla_{W_q^{lat}}), \tag{3}$$

where $\eta$ is the learning rate and $\sigma(\cdot)$ is the gradient clip function. Although Eq. 2 can provide the approximate gradients of $W_q^{lat}$, the estimation error cannot be ignored if we aim to further improve the performance of quantized networks. Considering that there are a number of layers and learnable parameters in modern neural networks, the optimization difficulty for learning accurate quantized models is still very large.

## 3.2 Low-bit Weight Searching

The parameters in a quantized neural network can be divided into two parts: the non-quantized parameters $W_f$ (*e.g.*, in batch normalization, fully connected layer), and the quantized convolutional

parameters $W_q$. The target of training the network is to find $W_f^*, W_q^*$ that minimizes

$$W_f^*, W_q^* = \underset{W_f \in \mathbb{R}, W_q \in \mathbb{V}}{\arg\min} \mathcal{L}(W_f, W_q), \tag{4}$$

where the $W_f$ and $W_q$ belong to different domains, which are real number $\mathbb{R}$ and discrete set $\mathbb{V}$, respectively.

To optimize modern neural networks, the stochastic gradient descent (SGD) strategy is usually utilized. However, the discrete variables $W_q$ are not feasible to be optimized by SGD. The previously described solution (Eq. 2- 3) by introducing the latent variables can be viewed as a heuristic solution that enables updating all the parameters in an end-to-end manner. However, the estimated and inaccurate gradients limit the upper bound of the performance. In contrast, inspired by gradient-based neural architecture search methods [46, 68, 24, 74, 12], we introduce the continuous relaxation strategy to search discrete weights.

Consider optimizing an $n$-dimensional discrete variable $W$ of size $(d_1, \ldots, d_n)$, where each element $w$ in $W$ is chosen from the $m$ discrete values $\mathbb{V} = \{v_1, v_2, \cdots, v_m\}$. A new auxiliary tensor $A \in \mathbb{R}^{m \times d_1 \times \cdots \times d_n}$ is created to learn the distribution of $W$, and the probability over $m$ discrete variables is computed according to the following formula,

$$P_i = \frac{\exp^{A_i/\tau}}{\sum_j \exp^{A_j/\tau}}, \quad i \in \{1, \cdots, m\}, \tag{5}$$

where $P_i$ is probability that the elements in $W$ belong to the $i$-th discrete value $v_i$, and $\tau$ is the temperature controlling the entropy of this system. The expectation of the quantized values for $W$ can be obtained:

$$W_c = \sum_i P_i \cdot v_i, \tag{6}$$

where the continuous state tensor $W_c$ is calculated according to the probability over all the discrete values. $W_c$ is used for convolution during training, and the process of Eq. 6 is differentiable and friendly for end-to-end training. Here we optimize the auxiliary tensor $A$ whose gradients can be accurately calculated so that we avoid the gradient estimation in previous works.

In the inference stage, the weights are quantized by selecting the discrete value with the maximum probability for each position. In formal, the quantized state weights are

$$W_q = \sum_i I_i \cdot v_i, \tag{7}$$

where $I = \text{onehot}(\arg\max_i(P_i)), \quad i \in \{1, \cdots, m\}$ indicates the one-hot vectors with the maximum probability.

For a convolution layer in neural networks, a four-dimensional weight tensor $W$ of size $M \times N \times D_K \times D_K$ is to be learned, where $M, N$, and $D_K$ are output channels, input channels, and kernel size, respectively. By using the proposed method to search for $q$-bit quantized weight $W$, the problem can be transformed into a special case of dimension $n = 4$ and discrete values $m = 2^q$ (Eq. 1). The tensor $A \in \mathbb{R}^{m \times M \times N \times D_K \times D_K}$ is constructed to learn the distribution over the discrete value set $\mathbb{V}$ according to Eq. 5. By using the continuous relaxation, the learning of Eq. 4 could be transformed to learn the tensor $A$,

$$W_f^*, A^* = \underset{W_f \in \mathbb{R}, A \in \mathbb{R}}{\arg\min} \mathcal{L}(W_f, A), \tag{8}$$

where the entire network is differentiable and can be optimized end-to-end with accurate gradients.

## 3.3 Optimization Details

We use the full-precision tensor $W_c$ for training and quantized tensor $W_q$ for inference. Although the continuous relaxation method solves the problem of inaccurate gradient during training, there is still a quantization gap after converting $W_c$ to $W_q$. This will lead to the mismatch between the $W_q$ and other parameters trained according to $W_c$, such as the statistics in the batch normalization layers.

**Algorithm 1** Training algorithm of SLB

---

**Input:** The network $\mathcal{N}$ constructed by convolution and state batch normalization, training iterations $I$, dataset $\mathcal{D}$, initialize temperature $\tau_s$, end temperature $\tau_e$ and temperature decay scheduler $\mathcal{T}$.
 1: **for** iter $i$ in $1, \ldots, I$ **do**
 2:     Update temperature parameter $\tau$ according to the temperature decay scheduler $\mathcal{T}$.
 3:     Get minibatch data $X$ and target $Y$ from dataset $\mathcal{D}$, calculate continuous weights $W_c$ and quantized weights $W_q$ of $\mathcal{N}$ according to the current temperature $\tau$.
 4:     The continuous state weights $W_c$ (Eq. 6) is computed and prediction $P = \mathcal{N}(X)$. The continuous state statistics in SBN are also updated.
 5:     Compute the discrete weights $W_q$ (Eq. 7) and the discrete state statistics in SBN layer are updated (Eq. 13).
 6:     The loss is calculated according to the prediction $P$ and target $Y$, and the back-propogated gradients are used to update auxiliary matrix $A$.
 7: **end for**
 8: Record quantized tensors $W_q$ and the discrete state statistics in SBN.
**Output:** A trained quantized neural network $\mathcal{N}^*$.

---

### 3.3.1 Reducing the Quantization Gap

In the proposed low-bit weight searching scheme, the continuous $W_c$ for training and the discrete $W_q$ for inference are not exactly the same. The quantization gap is introduced to measure the quantization error in the process of transforming the softmax distribution to the one-hot vector (Eq. 6- 7),

$$W_{gap} = W_q - W_c. \tag{9}$$

Fortunately, As stated in the temperature limit Theorem 1, the quantization gap $W_{gap}$ could be an infinitesimal quantity if the temperature $\tau$ is small enough.

**Theorem 1** ( Temperature Limit Theorem ). *Assuming $a \in \mathbb{R}^m$ is a vector in tensor $A$. If the temperature $\tau$ is gradually decreasing to zero, then the quantization gap is an infinitesimal quantity.*

*Proof.* The distribution $p$ is computed as,

$$p_i = \frac{\exp^{a_i/\tau}}{\sum_j \exp^{a_j/\tau}} = \frac{1}{\sum_j \exp^{(a_j - a_i)/\tau}}, \tag{10}$$

by gradually decreasing $\tau$ to zero, the index of the maximum value is $k = \arg\max_i p_i$, the quantization gap $w_{gap}$ is computed as,

$$
\begin{aligned}
\lim_{\tau \to 0} p_k &= 1, \quad \lim_{\tau \to 0} p_{i,\ i \neq k} = 0 \\
\lim_{\tau \to 0} w_{gap} &= v_k - \sum_i p_i \cdot v_i \\
&= (1 - p_k) \cdot v_k + \sum_{i,\ i \neq k} p_i \cdot v_i \\
&= 0.
\end{aligned}
\tag{11}
$$

$\square$

We propose the gradually decreasing temperature strategy during training so that $\lim_{\tau \to 0} W_c = W_q$. In particular, at the beginning of optimization, the temperature factor $\tau$ is high, and the distribution $P$ is relatively smooth. With the change of temperature $\tau$, the distribution $P$ becomes sharper. In this way, the quantization gap will gradually decrease as $\tau$ changes.

### 3.3.2 State Batch Normalization

With the gradually decreasing temperature strategy, the continuous $W_c$ will converge to the discrete $W_q$. Nevertheless, one problem that cannot be ignored is that the temperature cannot be decreased to a minimal value, *i.e.*, 0. If a neuron $w$'s probability of choosing its value to be a discrete value is

greater than a relatively high threshold (*e.g.*, $\max(p) > 0.999$), we may infer that the discrete value will not change as we continue to decrease the temperature and make the distribution sharper. Under this assumption, the quantized weights are well optimized and the only difference between training and inference is the statistics difference for batch normalization layers. Thus we proceed to develop a corresponding state batch normalization that acts as a bridge between a sharp softmax distribution and a one-hot distribution.

Batch Normalization (BN) [33] is a widely-used module for increasing the training stability of deep neural networks. The conventional BN for a given input feature $y$ can be written as,

$$\hat{y_i} = \frac{1}{\sigma_i}(y_i - \mu_i), \tag{12}$$

where $i$ is the index of channel and $\mu_i, \sigma_i$ are the mean and standard deviation values for the $i$-th channel, respectively. In the proposed method, $W_c$ is used for convolution during training, so the normalization is formulated as,

$$y_c = x \otimes W_c, \;\; \hat{y_{c,i}} = \frac{1}{\sigma_{c,i}}(y_{c,i} - \mu_{c,i}), \tag{13}$$

where $x$ is the input data, $\otimes$ is the convolution operation and $y_c$ is the output. $\mu_{c,i}$ and $\sigma_{c,i}$ are the mean value and standard deviation of the $i$-th channel in $y_c$.

Therefore, the statistics of the quantized weights $W_q$ does not match with the statistics $\sigma_c$ and $\mu_c$ of BN, which results in a precision decline in inference. To address this problem, we proposed the State Batch Normalization (SBN). To be specific, the SBN calculates two groups of statistics during training, one is for $y_c$ and the other is for $y_q$ where $y_q$ is the convolution output using quantized weights $W_q$. The normalization process of $y_q$ is

$$y_q = x \otimes W_q, \;\; \hat{y_{q,i}} = \frac{1}{\sigma_{q,i}}(y_{q,i} - \mu_{q,i}), \tag{14}$$

where $\mu_{q,i}$ and $\sigma_{q,i}$ are the mean and standard deviation of the $i$-th channel in $y_q$.

Both $\hat{y_c}$ and $\hat{y_q}$ have a mean of zero and a standard deviation of one. We make they share the same group of affine coefficients,

$$z_{c,i} = \gamma_i \hat{y_{c,i}} + \beta_i, \;\; z_{q,i} = \gamma_i \hat{y_{q,i}} + \beta_i \tag{15}$$

In this case, the proposed SBN eliminates the small quantization gap in the statistics for batch normalization layers.

### 3.3.3 Overall Algorithm

We view the training for quantized networks as a search problem, and utilize a differentiable method for learning the distribution of quantized weights. The strategies including gradually decreasing temperature and state batch normalization are proposed to eliminate the quantization gap. The overall training algorithm of searching for low-bit weights (SLB) is summarized in Algorithm 1.

## 4 Experiments

We examine our proposed method on quantized neural networks with different bitwidths. Following common practice in most works, we use the CIFAR-10 [37] and large scale ILSVRC2012 [9] recognition datasets to demonstrate the effectiveness of our method. For all the quantized neural networks, following previous methods [48, 76, 78], all convolution layers except for the first one are quantized. The low-bit set $\mathbb{V}$ is constructed by uniform distribution from $-1$ to $1$.

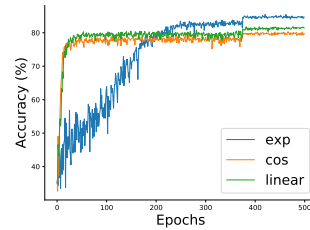

Figure 2: The diagram of different temperature scheduler.

### 4.1 Experiments on CIFAR-10

In this section, we first examine different temperature decay schedulers and then compare the results with other methods. We train the network for 500 epochs in

total and decay the learning rate by a factor of 10 at 350, 440, and 475 epochs. The matrix $A$ is initialized by kaiming initialization [26]. The quantization function for the activations is the same as DoReFa [78].

We test several different decay schedulers for temperature changing, including *sin*, *linear* and *exp*, which are widely used in adjusting the learning rate [49, 30, 49]. For convenience, we used $T$ in this section to represent $\frac{1}{\tau}$ in Eq. 5. The temperature $T$ is used to control the entropy of the system. Given the start temperature factor $T_s$, the end temperature $T_e$ and the total number of training iterations $I$. At iteration $i$, the temperature $T_t$ in different schedulers is calculated as follow,

$$T_{t,linear} = T_s + \frac{i}{I} \times (T_e - T_s), \ \ T_{t,sin} = T_s + \sin(\frac{i\pi}{2I}) \times (T_e - T_s), \ \ T_{t,exp} = T_s \times (\frac{T_e}{T_s})^{\frac{i}{I}}.$$

In Figure 2, we comprehensively compare different schedulers on binary ResNet20. We set $T_s = 0.01$ and $T_e = 10$. For *sin* and *linear* schedulers, the accuracies converge rapidly. However, some weights are not adequately trained, which results in relatively low accuracies. For the *exp* scheduler, the accuracies are much higher which indicates the weights converge to better local minima. In the following experiments, we adopt the *exp* scheduler for temperature adjusting.

Table 1: Results of ResNet20 on CIFAR-10.

| Methods | Bit-width (W/A) | Acc. (%) |
|---|---|---|
| FP32 [27] | 32/32 | 92.1 |
| DoReFa [78, 17] | 1/1 | 79.3 |
| DSQ [16] | 1/1 | 84.1 |
| SQ [17] | 1/1 | 84.1 |
| **SLB** | 1/1 | **85.5** |
| LQ-Net [76] | 1/2 | 88.4 |
| **SLB** | 1/2 | **89.5** |
| **SLB** | 1/4 | **90.3** |
| **SLB** | 1/8 | **90.5** |
| LQ-Net [76] | 1/32 | 90.1 |
| DSQ [16] | 1/32 | 90.2 |
| **SLB** | 1/32 | **90.6** |
| LQ-Net [76] | 2/2 | 90.2 |
| **SLB** | 2/2 | **90.6** |
| **SLB** | 2/4 | **91.3** |
| **SLB** | 2/8 | **91.7** |
| **SLB** | 2/32 | **92.0** |
| **SLB** | 4/4 | **91.6** |
| **SLB** | 4/8 | **91.8** |
| **SLB** | 4/32 | **92.1** |

Table 2: Results of VGG-Small on CIFAR-10.

| Methods | Bit-width (W/A) | Acc. (%) |
|---|---|---|
| FP32 [58] | 32/32 | 94.1 |
| BNN [32] | 1/1 | 89.9 |
| XNORNet [52] | 1/1 | 89.8 |
| DoReFa [78, 17] | 1/1 | 90.2 |
| SQ [17] | 1/1 | 91.7 |
| **SLB** | 1/1 | **92.0** |
| LQ-Net [76] | 1/2 | 93.4 |
| **SLB** | 1/2 | **93.4** |
| **SLB** | 1/4 | **93.5** |
| **SLB** | 1/8 | **93.8** |
| LQ-Net [76] | 1/32 | 93.5 |
| **SLB** | 1/32 | **93.8** |
| LQ-Net [76] | 2/2 | 93.5 |
| **SLB** | 2/2 | **93.5** |
| **SLB** | 2/4 | **93.9** |
| **SLB** | 2/8 | **94.0** |
| **SLB** | 2/32 | **94.0** |
| **SLB** | 4/4 | **93.8** |
| **SLB** | 4/8 | **94.0** |
| **SLB** | 4/32 | **94.1** |

We compare our results with other SOTA quantization methods including BNN, XNORNet, DoReFa, DSQ, SQ, and LQ-Net, on two different network architectures, *i.e.*, VGG-Small [58] and ResNet20 [27]. As shown in Table 1 and Table 2, our method outperforms other methods and achieves state-of-the-art results on different bit-widths.

## 4.2 Experiments on ILSVRC2012

In the ILSVRC2012 classification experiments, we use ResNet18 [27, 48, 56, 57] as our backbone and compare the results with other state-of-the-art methods. The learning rate starts from 1e-3, weight decay is set to 0, and Adam optimizer is used to update parameters. Table 3 presents the results of our method and a number of other quantization methods.

The binary neural network is the most potential method because the xnor and bitcount operations are relatively efficient. Our proposed method on the binary case achieves state-of-the-art accuracy with a

Table 3: Overall comparison of quantized ResNet18 on ILSVRC2012 large scale classification dataset. 'W/A' denotes the bitwidth of weights and activations, respectively.

| Methods | Bit-width (W/A) | Top-1 (%) | Top-5 (%) |
|---|---|---|---|
| FP32 [27] | 32/32 | 69.3 | 89.2 |
| BNN [32] | 1/1 | 42.2 | 67.1 |
| ABCNet [43] | 1/1 | 42.7 | 67.6 |
| XNORNet [52] | 1/1 | 51.2 | 73.2 |
| BiRealNet [48] | 1/1 | 56.4 | 79.5 |
| PCNN [18] | 1/1 | 57.3 | 80.0 |
| SQ [17] | 1/1 | 53.6 | 75.3 |
| ResNetE [2] | 1/1 | 58.1 | 80.6 |
| BONN [19] | 1/1 | 59.3 | 81.6 |
| **SLB** | 1/1 | **61.3** | **83.1** |
| SLB (w/o SBN) | 1/1 | 61.0 | 82.9 |
| DoReFa [78] | 1/2 | 53.4 | - |
| LQ-Net [76] | 1/2 | 62.6 | 84.3 |
| HWGQ [5] | 1/2 | 59.6 | 82.2 |
| TBN [63] | 1/2 | 55.6 | 79.0 |
| HWGQ [5] | 1/2 | 59.6 | 82.2 |
| **SLB** | 1/2 | **64.8** | **85.5** |
| DoReFa [78] | 1/4 | 59.2 | - |
| **SLB** | 1/4 | **66.0** | **86.4** |
| SYQ [13] | 1/8 | 62.9 | 84.6 |
| **SLB** | 1/8 | **66.2** | **86.5** |

| Methods | Bit-width (W/A) | Top-1 (%) | Top-5 (%) |
|---|---|---|---|
| FP32 [27] | 32/32 | 69.3 | 89.2 |
| BWN [52] | 1/32 | 60.8 | 83.0 |
| DSQ [16] | 1/32 | 63.7 | - |
| SQ [17] | 1/32 | 66.5 | 87.3 |
| **SLB** | 1/32 | **67.1** | **87.2** |
| PACT [7] | 2/2 | 64.4 | - |
| LQ-Net [76] | 2/2 | 64.9 | 85.9 |
| DSQ [16] | 2/2 | 65.2 | - |
| **SLB** | 2/2 | **66.1** | **86.3** |
| **SLB** | 2/4 | **67.5** | **87.4** |
| SYQ [13] | 2/8 | 67.7 | 87.8 |
| **SLB** | 2/8 | **68.2** | **87.7** |
| TTQ [79] | 2/32 | 66.6 | 87.2 |
| LQ-Net [76] | 2/32 | 68.0 | 88.0 |
| **SLB** | 2/32 | **68.4** | **88.1** |

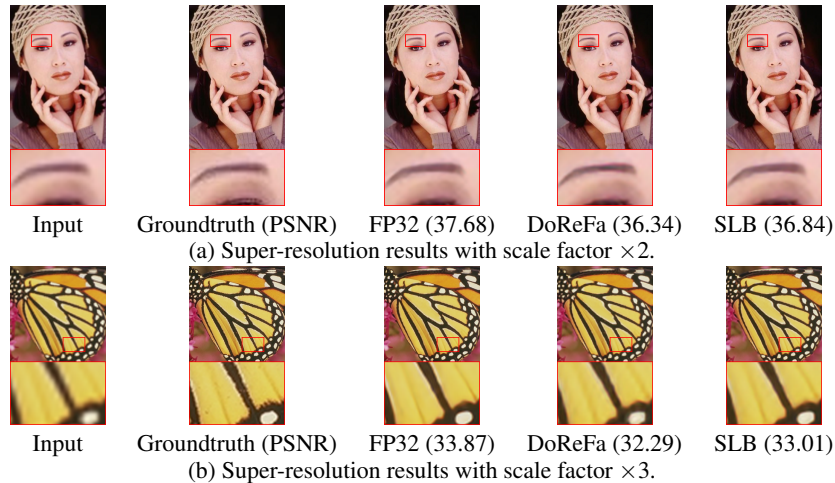

Input    Groundtruth (PSNR)    FP32 (37.68)    DoReFa (36.34)    SLB (36.84)
(a) Super-resolution results with scale factor ×2.

Input    Groundtruth (PSNR)    FP32 (33.87)    DoReFa (32.29)    SLB (33.01)
(b) Super-resolution results with scale factor ×3.

Figure 3: Super-resolution results on Set5 dataset with different scale factors.

Top-1 accuracy of 61.3% . When using more bits on the activations, the Top-1 accuracies gradually increase. Our method consistently outperforms other methods by a significant margin. By adding the bitwidth of weights, the proposed SLB also achieves state-of-the-art accuracies on different settings.

### 4.2.1 The Impact on State Batch Normalization

We further verify the importance of state batch normalization by removing it in binary ResNet18. In particular, the discrete weights and discrete state statistics are not calculated during training, and the continuous state statistics are used after training. Because of the quantization gap, the SLB (w/o SBN) in Table 3 decreases the Top-1 accuracy by 0.3%. This indicates the essentiality to maintain two groups of statistics, which are estimated by the continuous state outputs and discrete state outputs, respectively.

### 4.3 Experiment on Super Resolution

To verify the generalization of the proposed method, we apply SLB to the image super-resolution task. The typical model VDSR [36] is selected as the baseline model. Following the original paper, we use the 291 images as in [55] for training and test on Set5 dataset [3]. Each image in the training data is split into patches and augmented with rotation or flip. The baseline model and the binarized models using DoReFa and SLB algorithms are trained with the same setting as in [36].

The models are compared at different scale factors, *i.e.* $\times 2$ and $\times 3$. From the results in Figure 3, we can see that SLB achieves much higher PSNR than DoReFa at both $\times 2$ and $\times 3$ scale factors. In Figure 3(a), the eyebrow in DoReFa's result is somewhat red while the result of SLB is normal and closer to the raw image. In Figure 3(b), the texture on the wing in DoReFa's result is blurry and hard to see clearly. SLB could well recover the texture and performs close to the full-precision model.

## 5 Conclusions

In this paper, we use the continuous relaxation strategy which addresses the gradient mismatch problem. To learn the discrete convolution kernels, an auxiliary probability matrix is constructed to learn the distribution for each weight from soft to hard, and the gradients are calculated to update the distribution. The state batch normalization is also proposed to minimize the gap between continuous state outputs and discrete state outputs. Our SLB can be applied to optimize quantized networks on a number of bitwidth cases and achieved state-of-the-art performance, which demonstrates the effectiveness of our method.

## Acknowledgement

This work is supported by National Natural Science Foundation of China under Grant No. 61876007, Australian Research Council under Project DE180101438.

## Broader Impact

Compared with full-precision networks, the quantized neural networks have the advantages of small model size, fast inference speed, low energy cost and efficient runtime memory occupation. The methods for training high-precision quantized neural networks help the deployment of computer vision models on mobile devices. Our proposed searching scheme provides a novel and feasible method for training high-precision quantized networks.

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
