[Supplementary Material]

# Searching for Low-Bit Weights in Quantized Neural Networks

**Zhaohui Yang**[1,2]**, Yunhe Wang**[2]**, Kai Han**[2]**, Chunjing Xu**[2]**,**
**Chao Xu**[1]**, Dacheng Tao**[3,*] **Chang Xu**[3]
[1] Key Lab of Machine Perception (MOE), Dept. of Machine Intelligence, Peking University.
[2] Noah's Ark Lab, Huawei Technologies.
[3] School of Computer Science, Faculty of Engineering, University of Sydney.
`zhaohuiyang@pku.edu.cn; {yunhe.wang,kai.han,xuchunjing}@huawei.com`
`xuchao@cis.pku.edu.cn; {dacheng.tao, c.xu}@sydney.edu.au`

## A  Visualization of Distribution

We visualize the probability $P$ of the first binary convolution layer in ResNet20 (1/1) in Figure 1. The different curves in the diagram represent the percentage of weights, where the max probability is higher than the given thresh. As can be seen from the figure, the weights converge gradually. As the temperature increases, the distribution becomes much sharper, and the easily optimized neurons first show the tendency toward discrete values.

Figure 1: The distribution of the first convolution layer in binary ResNet20.

---

[*]Corresponding Author.