[Reviews · NeurIPS 2020]

Review 1

Summary and Contributions: The papper introduce a new differentiable method for searching discrete low-bit weights in quantized networks. A weight searching algorithm is used to avoid the non-differentiable problem of quantization functions. To further minimize the performance gap after searching, a temperature factor and the state batch normalization are proposed for seeking consistency in both training and testing. The experiments on both high-level and low-level tasks have demonstrated effectiveness of the proposed methods.

Strengths: The proposed weight searching method can replace the conventional methods by utilizing approximate gradients to optimize the latent weights. The searching approach is novel and interesting. The formulation on how to optimize the distribution is clear. This paper is good on the discrete weights that could also be learned together with the distribution, which may motivate followers. The experimental results are convincing with the state-of-the-art results. The experiments include various networks and tasks, and show promising results under different bit-width settings.

Weaknesses: This paper introduce a novel weight searching algorithm and show its effectiveness. but I have several questions and suggestions about this paper: There are several Typos needed to be fixed, e.g. Alg. 1, 5, '13' should be 'Eq. 13'. The stage batch normalization is motivated to handle the quantization gap based on a threshold, and the author assumes the distribution will not change. Is the distribution irreversible? Please discuss the relationship between the proposed method with some more recent works, such as Latent Weights Do Not Exist: Rethinking Binarized Neural. NeurIPS2019

Correctness: yes

Clarity: yes, well done

Relation to Prior Work: yes, related work is well done too.

Reproducibility: Yes

Additional Feedback:


Review 2

Summary and Contributions: In this manuscript, authors consider the problem that non-differentiable quantization methods increase the optimization difficulty of quantized networks. By representing quantized model weights as the expectation of a parameterized probability distribution, end-to-end gradient-based optimization can be adopted. The quantization gap, between floating-point and quantized model weights, is reduced by dynamically adjusting the softmax temperature, and state batch normalization is proposed to account for the inconsistent statistics between FP32 and discrete model weights.

Strengths: 1. The idea of representing discrete weights as the expectation of a parameterized probability distribution is novel and reasonable, and indeed resolves the non-differentiable issue of discrete weights. The quantization gap is gradually reduced by adjust the temperature factor in the softmax function. 2.By maintaining two groups of statistics for full-precision and quantized weights during training, the state batch normalization method explicitly consider the difference between a sharp softmax distribution and a one-hot distribution, which improves the final inference accuracy with quantized weights.

Weaknesses: 1.The decreasing temperature strategy. This plays an important role in balancing the optimization efficiency and reducing the quantization gap. Is it possible to derive some theoretical guidelines for designing the decreasing scheduler? The three schedulers (“exp”, “cos”, and “linear”) used in the experiments seem quite heuristic, and the best one, “exp”, only outperforms the other two after 200 epochs and ends up with a much better solution. 2.If the major dilemma is to avoid local minimum while reducing the quantization gap, is it possible to adopt the SGD-R [1] like schedule for adjusting the temperature factor? The quantization gap should be quite small at the end of each period, and local minimum can be avoided with each warm restart. 3.The state batch normalization method requires maintaining two groups of statistics during the whole training process. For each mini-batch, this requires two forward passes, one with FP32 weight and the other with quantized weights. This introduces extra computation overhead. Is it possible to only compute statistics for quantized weights after the training process is completed? 4.Do you have SLB w/o SBN results on CIFAR-10? It would be easier to analyze which part contributes more to the final improvement, if such results are available. 5.For the CIFAR-10 dataset, 500 epochs are used to train the quantized network. This is significantly larger than the setting in some of baseline methods. For instance, LQ-net only uses 200 epochs. In this case, the comparison may not be quite fair. Does the proposed method really require so much epochs (maybe due to the optimization efficiency), or it can also be well trained using much fewer epochs? [1] Ilya Loshchilov and Frank Hutter. SGDR: Stochastic Gradient Descent with Warm Restarts. ICLR 2017.

Correctness: Yes, the proposed method is reasonable and empirical evaluation results have verified its effectiveness on several tasks and benchmark datasets.

Clarity: Yes

Relation to Prior Work: Yes

Reproducibility: Yes

Additional Feedback: Please address issues listed in the “Weaknesses” section. ---- Authors have resolved most of my previous concerns in their rebuttal, especially the temperature decaying strategy and calibrating BN statistics. It would be nice to include results achieved with the SGD-R based temperature scheduler, as mentioned in the rebuttal. I would like to raise my score to 6 and recommend acceptance.


Review 3

Summary and Contributions: This paper proposes to train a low-precision quantized network by searching over the possible values for the network parameters. The proposed method achieves state-of-the-art results on the CIFAR-10 and ImageNet datasets.

Strengths: This paper proposes a novel way to find the quantized weights' discrete value, which addresses the imprecise gradient problem in the STE-based methods. Moreover, the authors propose a temperature scheduling method to minimize the difference between the weights during training and inference.

Weaknesses: The proposed method consumes more device memory as it requires storing the probability distribution of each weight. In the case of a 4-bit weight, the proposed method requires 16 times larger device memory to store the weights than STE-based approaches. As the size of the weights becomes larger, the total training time is assumed to be longer as well. Moreover, as stated in the CIFAR-10 experiment, the proposed method takes 500 epochs. It is not clear whether the proposed would require more epochs to converge than STE-based methods.

Correctness: The proposed method is sound and the empirical methodology is correct. However, some details about the empirical study are missing.

Clarity: The paper is clear and well written.

Relation to Prior Work: The related works are discussed clearly.

Reproducibility: No

Additional Feedback: Can you please provide the rationale of setting the low-bit set V to be in the range of -1 to 1? When the bit-width is higher than 1, how does this choice affect the accuracy of the quantized network? The reported results on the CIFAR-10 datasets are using 500 epochs whereas other methods could take much less number of epochs. For example, the LQ-Net uses 200 eopchs for the CIFAR-10 dataset. Moreover, the hyperparameter of training is missing for the ImageNet experiments. For both the CIFAR-10 and ImageNet experiments, can you comment on the proposed method's total training time? For the CIFAR-10 experiments, can you please show the error bar of the experiments? Besides, the proposed method is superior to the STE-based methods when the precision of the weights is low. Can you also comment on when the proposed method is significantly better than the existing STE-based alternatives? ------------ after rebuttal ------------- The authors have responded to most of my questions in their rebuttal. Although the improvement over the STE-based method for quantization with multiple bits isn't very impressive, the paper does provide an interesting solution for weight quantization. I would like to keep my score to 6 and recommend acceptance.


Review 4

Summary and Contributions: This paper introduce a method which uses an auxiliary trainable network to build a quantized neural network. To minimize the gap between the two networks, a quantization loss is proposed to minimize it. Two datasets are used to valudate the proposed method. The similar framework has been proposed in IJCAI 2019.

Strengths: The idea is interesting and the topic is very important for resource-efficient computation.

Weaknesses: 1) The similar idea of learning an auxiliary differentiable network has also been introduced in the following paper. The main difference of this paper to the following reference is that multiple bits are learned for each code in this paper while, undoubtedly, binary weights and representations will be more cost-efficient. More importantly, authors did not discuss this similar reference. [1] Binarized Neural Networks for Resource-Efficient Hashing with Minimizing Quantization Loss. IJCAI, 2019 2) I am very confused with the EQ. (6) which is most important equation in this paper. According to EQ. (1), The values $v$ are discrete numbers while $p$ is probability that the elements in $W$ belong to the $i$-th discrete value. Take one example, if we assume $v\in {1,2,3,4} and $p={0.4,0.2,0.2,0.2}$, the results would be 2.2 which is more close to 2. So, authors should carefully check and correct this equation or add more explanations. 3) The experimental setting is not clear. Abalation study is missed and thus, it is not clear which part is more important. -------------after reading the rebuttal I have read the rebuttal and the review comments from others. Although the clarity of this paper should be carefully improved, the novelty issue, particularly for the differences between this paper and the reference I mentioned, has been well addressed by the authors. After carefully re-read the paper, I found that the sign function which is non-differentiable is not used in this paper, so the following optimization will be much easier than those have sign function and the paper seems to be more special than what I orginally thought. Moreover, considering such a weight search method can obtain SOTA performance on ImageNet, I would like to increase my score to 5. I suggest the authors include more recent related works into the final version and highlight the novelty accordingly.

Correctness: Some equations are hard to understand. The experiment is not clear.

Clarity: Equations need to be explained in detial. Some of them (Eq. 6) need to be corrected.

Relation to Prior Work: The main drawback is that this paper didn't discuss the highly-similar work.

Reproducibility: No

Additional Feedback:

[Author Response · NeurIPS 2020]

We sincerely thank four reviewers for the valuable comments. The training epoch issue is concerned by Reviewer #2,
#3. The ablation study issue is concerned by Reviewer #2, #4. We answer these issues first.
**Epochs.** The LQ-Net trains 400 epochs on CIFAR-10 in total [r1] (the $max\_epoch$ parameter). We used 500 epochs to
ensure the maximum accuracy is achieved. We found 400 (same with LQ-Net) epochs on the CIFAR-10 dataset could
derive the same accuracies.
**Ablation study.** We have already conducted the ablation study on the temperature in Sec. 4.1 (Fig. 1) and the ablation
study on SBN in Sec. 4.2.1 (Tab. 3). For the ablation of SLB w/o SBN on CIFAR-10, the accuracies of SLB-VGG-Small
(W/A = {1/1, 2/2, 4/4}) w/o SBN on CIFAR-10 are 91.9% (-0.1%), 93.2% (-0.3%), and 93.6% (-0.2%). The temperature
scheduler brings 4-6% accuracy improvement on CIFAR-10 . The temperature scheduler contributes more to the final
performance compared to the SBN.
**For Reviewer #1**
**Typos.** Thanks. We fill fix the typos in the updated version and proofread the paper to make it more readible.
**Distribution.** As shown in Supp Fig. 1, the distribution becomes sharper during training. The distribution is not
irreversible, and the gap between the soft distribution and hard representation becomes small.
**Related works.** The LWDNE (NeurIPS2019) introduces a novel optimizer for optimizing binary networks. Our SLB
uses the SGD and Adam optimizer to optimize networks with several bit-widths. We will cite and discuss with LWDNE.
**For Reviewer #2**
**Temperature.** Thanks for the nice suggestions. We adopt these three schedulers in our experiments, because of their
proven performance in a number of tasks, e.g., NAS (*e.g.*, exp scheduler in FBNet, CVPR2019), network optimization
(*e.g.*, cos scheduler in DARTS, ICLR2019) and curriculum learning (Fig. 1 in [r2]). This paper aims for a new
framework of searching low-bit weight. The designing and theoretical analysis of the schedulers are interesting but out
of our scope. This will be nice follow ups to our current studies.
**SGD-R.** Thanks for this insightful comment. We tried the SGD-R scheduler to train VGG-Small (1/1) on the CIFAR-10
dataset. We warmup the temperature for three times. The accuracy is 92.3% which is 0.3% higher than the result trained
w/o SGD-R. We will discuss this in the updated version.
**The BN calibration.** This step accelerates training by tuning BN for a few epochs. Calibrating the BN statistics after
training could be efficient but would hurt the accuracy. However, the network at the last epoch is always not the best
one. The second reason is that we could not observe the accuracies in the early/middle epochs because of the large
quantization gap.

Table 1: The GPU consumption and time cost of VGG-Small on CIFAR-10.

| Metric | Network | Dataset | STE | SLB(1/1) | SLB(2/2) | SLB(4/4) |
|--------|---------|---------|-----|----------|----------|----------|
| GPU (G) | VGG-Small | CIFAR-10 | 3.1 | 3.1 | 3.2 | 4.0 |
| Time (h) | VGG-Small | CIFAR-10 | 4.7 | 6.4 | 7.5 | 9.2 |
| Time (h) | ResNet18 | ImageNet | 32 | 33 | 38 | 59 |

**For Reviewer #3**
**GPU memory consumption.** The memory consumption
of VGG-Small trained on CIFAR-10 dataset is detailed
in Tab. 1. The batch size is 100. In the case of 4-bit weight, only the weights require 16 times larger. During training,
the GPU consumption on feature maps is much more than the GPU consumption on weights. The SLB-VGG-Small
(4/4) uses around 30% extra memory compared to STE-VGG-Small. More GPU memory consumption is only required
during training and our method do not increase the inference cost.
**Low-bit set range.** We follow DoReFa and set the value range to $[-1, 1]$. When the bit-wise is higher than 1, the
accuracy might be higher because the information loss is small. For example, PACT [r3] makes the clipping value
learnable and achieves higher accuracies.
**Training time.** We use 1 V100 to train networks on CIFAR-10 and 8 V100 to train networks on ImageNet. The time
cost of VGG-Small on CIFAR-10 and ResNet18 on ImageNet is shown in Tab. 1.
**Experimental details.** We use Adam optimizer to train the architectures on the ImageNet dataset for 120 epochs in
total with a batch size of 256, learning rate starts from 1e-3 and decays by a factor 10 at epoch 70, 90 and 110. Weight
decay is 0. The temperature starts from 0.01, ends at 10.0, and changes by the exp scheduler.
**CIFAR-10 error bar.** We ran SLB-VGG-Small (W/A = {1/1, 2/2, 4/4}) (Tab. 2) for five times and the results are
(mean±std format) 92.1±0.07, 93.5±0.08, and 93.8±0.05, respectively. The results are relatively steady.
**Comparison with STE.** The main problem of STE is that the gradients calculated relative to the quantized values are
used to update the latent variables (Eq. 2). Our proposed method performs significantly better than STE-based methods
on the condition that the differences between $W_q$ and $W_q^{lat}$ is large.
**For Reviewer #4**
**Related works.** Thanks for pointing out this issue. We will cite [r4] and discuss the differences between them.
Admittedly, [r4] is an excellent work to explore differentiable quantization. However, the conventional sign function
(Eq. 2 in [r4]) is still used to binarize weight parameters, which is still non-differentiable and Eq. 4 in [r4] can be seen
as a kind of regularizer on the weight matrix $W_l$. In contrast, we utilize Eq. 6 in our main body to represent quantized
weights during the training and do not use the sign function. For the binary case, each weight is represented by {+1,
-1} at the same time, and both the two values and the probability $P$ will be used to calculate the output features. By
exploiting the proposed method, we can obtain state-of-the-art performance on benchmark network architectures on
ImageNet using low-bit weights. Moreover, [r4] can only work for binary codes (see Eq. (4) in [r4]) and focus on
hashing comparison experiments. Instead, ours is applicable for multiple bits and we mostly consider quantization
comparison methods. This is exactly because we have a complete different algorithmic formulation.
**Eq. 6.** Eq. 6 is the expectation of the quantized values during training, while $W_q$ in Eq. 7 is the finalized quantized
values for inference. Given the example by #R4, we will take 2.2 for training but 1 for inference. Notably in fact,
this gap will not be so large, as we will sharpen the distribution by adjusting the temperature during training (see our
Theorem 1).

[1] https://github.com/microsoft/LQ-Nets/blob/master/cifar10-vgg-small.py#L151
[2] Dynamic Curriculum Learning for Imbalanced Data Classification. ICCV2019
[3] PACT: Parameterized Clipping Activation for Quantized Neural Networks. Arxiv
[4] Binarized Neural Networks for Resource-Efficient Hashing with Minimizing Quantization Loss. IJCAI2019


[Meta-Review · NeurIPS 2020]

The paper proposes a novel end-to-end gradient-based optimization for searching discrete low-bit weights in quantized networks. After reading the reviews, rebuttal, and the discussion among reviewers the paper clearly is recognized as novel and well executed. I would encourage the authors to further improve their work by better clarifying the decay strategy for the temperature in the camera ready and to add a comparison with SGD-R scheduling as pointed out by one of the reviewers. It would be also nice to have a mention on how the proposed approach relates to Latent Weights Do Not Exist: Rethinking Binarized Neural. NeurIPS2019 as pointed by R1.